# Dietary Effects on the Gut Phageome

**DOI:** 10.3390/ijms25168690

**Published:** 2024-08-09

**Authors:** Andrea Howard, Amanda Carroll-Portillo, Joe Alcock, Henry C. Lin

**Affiliations:** 1School of Medicine, University of New Mexico, Albuquerque, NM 87131, USA; anhoward@salud.unm.edu; 2Division of Gastroenterology and Hepatology, University of New Mexico, Albuquerque, NM 87131, USA; 3Department of Emergency Medicine, University of New Mexico, Albuquerque, NM 87131, USA; joalcock@salud.unm.edu; 4Medicine Service, New Mexico VA Health Care System, Albuquerque, NM 87108, USA

**Keywords:** microbiome, bacteriophage, phageome, diet, dysbiosis

## Abstract

As knowledge of the gut microbiome has expanded our understanding of the symbiotic and dysbiotic relationships between the human host and its microbial constituents, the influence of gastrointestinal (GI) microbes both locally and beyond the intestine has become evident. Shifts in bacterial populations have now been associated with several conditions including Crohn’s disease (CD), Ulcerative Colitis (UC), irritable bowel syndrome (IBS), Alzheimer’s disease, Parkinson’s Disease, liver diseases, obesity, metabolic syndrome, anxiety, depression, and cancers. As the bacteria in our gut thrive on the food we eat, diet plays a critical role in the functional aspects of our gut microbiome, influencing not only health but also the development of disease. While the bacterial microbiome in the context of disease is well studied, the associated gut phageome—bacteriophages living amongst and within our bacterial microbiome—is less well understood. With growing evidence that fluctuations in the phageome also correlate with dysbiosis, how diet influences this population needs to be better understood. This review surveys the current understanding of the effects of diet on the gut phageome.

## 1. Introduction

Bacteriophages (phages), viruses that infect bacteria, are key to the function and steady state of their matched bacteria. With a variety of life cycles including lytic, lysogenic, pseudolysogenic, and chronic (reviewed in [1]), it is possible for phages to dynamically regulate bacterial populations in response to environmental stimuli including diet. Dogma for phages posits that bacterial hosts are phage-specific, with each phage able to infect only a specific species of bacteria. However, recent examples of phages with broad host specificity challenge this premise. Modi et al. [2] demonstrated in vivo that exposure to certain stressors, such as antibiotics, made bacteria more susceptible to a variety of phages. Experimentally, using Appelman’s protocol in an in vitro setting has been shown to broaden host range through the phage recombination that occurs during iterative passages [3,4,5].

Phages are ubiquitous in terrestrial and marine environments, with density as high as 10^10^ per liter seawater, outnumbering their bacterial hosts 10:1 and driving microbial ecology through predatory interactions [6,7,8,9,10]. Comparatively, bacteriophages of the healthy intestinal tract are predominately prophages (maintain themselves within their bacterial hosts without lysis) [11,12] and exist at a ratio of 1:1 with their bacterial hosts [13,14,15]. They too are reported to impact the bacterial populations of the gastrointestinal (GI) system, although through different mechanisms than their environmental counterparts [16,17,18,19]. Recent advances in technologies for phage enrichment/identification along with more complete phage databases have allowed for a better understanding of the diversity within GI phage populations [20,21,22], and given that bacteria within the gut number in the order of 10^7^ to 10^14^ depending on location [23], there are plenty of hosts for bacteriophages to infect.

The addition of exogenous phages or manipulating the resident GI phage population to modify disease conditions is of particular interest in medicine. Phage therapy—administering exogenous phages for treatment of bacterial infections or overgrowth of colonizing pathogens—has proven successful in limited instances including treatment of *Clostridium* [24,25,26], *Pseudomonas* [27,28], and *Staphylococcus* [29] infections. The resident phage population can also regulate the bacterial microbiome and has a role in the transmission of genes involved in antibiotic resistance and bacterial toxins making phages an attractive target for therapeutic use [30,31,32].

Until recently, the greatest challenge with research on the GI phageome was the lack of information on the phages comprising this population. Phage databases were of limited use as most sequences were derived from environmental samples. However, with the advent of improved sequencing techniques, the viral “dark matter” in the human gut has become much more visible leading to an explosion of information regarding the constituents of the GI phageome [20,21,33]. The human phageome is temporally and spatially diverse [34] with phages first being found a few days after birth [35,36] with periods of richness during both infancy (0–3 years) and adulthood (18–65 years) [37]. Fluctuations in phage populations are associated with changes in diet and/or the bacterial population such as those occurring when the calorie source transitions from breast milk to solid foods during concurrent age-related shifts in the GI microbial community [38,39,40,41]. For example, individuals from birth to about 2.5 years of age will shift from no phage at birth to *Caudoviricetes* dominance until about 2.5 years, followed by a phageome shift to *Microviridae* dominance [22]. In contrast, the adult phageome (18–65 years) is stable, individualized, and dominated by DNA phages with most phages found in class *Caudoviricetes* (dsDNA, tailed) (families *Autographiviridae*, *Ackermannviridae*, *Chaseviridae*, *Demerecviridae*, *Drexlerviridae*, and *Herelleviridae*) [42,43] and families *Microviridae* (ssDNA, icosahedral) and *Inoviridae* (ssDNA, filamentous) [21,44,45,46,47]. While a core virome, similar to a core bacteriome, was suggested early on in all humans [48], and more recently, a comparison based on much larger patient samples demonstrated that the phageome is highly variable between individuals with variations linked to geography, lifestyle, and diet [37,46,49].

## 2. Activation of Phages by External Stimuli

Within the GI tract, a majority of phages exist in a lysogenic or pseudolysogenic state [12,50]. The number of integrases—markers of lysogeny phenotype—found within the sequenced virome suggests that, in a healthy phageome, most phages are quiescent or possess low lytic capabilities with some phages released into the extracellular milieu without lysis of the host such as the crAssphages and *Inoviridae* ([51] and reviewed in [35,36]). However, the observed phage phenotype may be tied to external stimuli as a number of factors have been identified as capable of triggering a switch from a lysogenic/pseudolysogenic life cycle to a lytic life cycle ([52] and reviewed in [53,54]). With this switch, the external—or “free”—phages become available to not only affect changes in bacterial abundances but to also regulate mammalian cellular responses from the endothelium or underlying immune cells ([55]; reviewed in [56,57,58]). External stimuli capable of inducing prophage may include changes in dietary elements, discussed further herein, soluble metabolites from the bacteria within the microbiome such as autoinducer types and concentrations [59,60,61], or environmental triggers such as oxygen [53,62,63,64,65]. Of all these external factors, diet has the potential to play the largest role.

## 3. Dietary Effects on the Gut Phageome

Dietary changes have rapid and powerful effects on the microbial ecology of the GI system [66,67,68]. The diet provides a source of carbon, protein, and growth-limiting micronutrients to gut microbes. These nutrients directly affect which microbial populations gain a competitive advantage, and indirectly influence the concentration of metabolites and quorum-sensing molecules that dominate the space. These factors determine the host population available for bacteriophages and can also inhibit or trigger the activation of prophages. Thus, for each dietary change that is made, there are two possible ways in which the phageome may regulate the microbial constituents of the microbiome: (1) the dietary shift causes changes in the microbial population, which in turn causes observable shifts within the phageome population; or (2) dietary shifts results in changes in the phageome population either through induction of prophage or activation/inhibition of some bacteriophage activity, and this phageome has direct effects on the bacterial microbiome constituents.

### 3.1. Western Diet—High Fat

The Western Diet (WD), characterized by high fat, high sugar, and low fiber, is the most common diet studied in relation to its effects on GI microbial populations with a significant role in altering not only the bacteriome, but the phageome as well. To better understand the role of WD on microbiome regulation, the contribution of each of the dietary components (fat, fiber, and sugar) has been assessed. Phage abundance has been found to increase with the implementation of a high-fat diet. In 2016, Kim et al. [63] assessed the effects of a combinatorial high-fat, high-sucrose (HFHS) diet on the distribution and phenotype of both bacterial and phage constituents of the colonic microbiome. They found HFHS-fed mice lost spatial variation between the mucosal and luminal phage and bacterial communities within the colon that existed in their low-fat diet-fed lean mouse counterparts. Additionally, Kim et al. reported an overall increased number of *Caudoviricetes* phages, but more so in the mucosa than the lumen. These phage increases were proposed to occur through prophage induction of bacteria coming mainly from *Bacteroidia*, *Bacilli*, and *Negativicutes* classes based on the prediction of putative bacterial hosts of viral sequences with homologs in sequencing databases. Schulfer et al. [69] followed up in 2020 focusing solely on the contribution of a high-fat diet (HFD) and found that mice fed a HFD had a significant decrease in *Caudoviricetes* (specifically in phages previously identified as *Siphoviridae*) with a concurrent increase in *Microviridae* and decrease in the integrase proportion in their fecal phageome. As integrases are important markers for the presence of lysogenic bacteriophages, a decrease in number implies an individualistic systemic shift from temperate bacteriophages to lytic phages. HFD-fed mice also showed an increase in temperate phages associated with Bacteroidota, a significant decrease in bacterial alpha diversity, and bacterial abundance change from Bacteroidota dominance to Bacillota dominance. Viral sequencing in these mice demonstrated no change to alpha diversity but a significant change in beta diversity. Other studies have shown that HFD causes a decrease in phage diversity more quickly than bacterial diversity [70,71]. Higgins et al. [71] found a reduction in *Caudoviricetes* phages (previously identified as *Podoviridae* and *Myoviridae*) in HFD-fed mice as early as 2 days and 8 weeks, respectively, as compared to mice given a standard diet. Hallowell et al. [70] found that administration of HFD to pigs resulted in immediate and significant decreases in *Streptococcus* phage *Sfi21dt1*.

### 3.2. Western Diet—Low Fiber

Low fiber is another feature of the WD requiring bacteria within the microbiome to find equivalent nutrient sources, such as through glycan foraging. In vivo, glycan foraging primarily occurs when bacteria utilize mucin *O*-glycans as a substrate [72], and phages may play a role in regulating the bacterial genes mediating this process. In low-fiber states, glycan-foraging bacteria erode mucus that protects the gut epithelium, thereby promoting intestinal permeability and susceptibility to microperforation, inflammation, and infection [73]. A study of mice given a fiber-free diet resulted in a substantially thinned mucus (5–6 times) as compared to their fiber-rich counterparts [73]. Furthermore, a long-term, low-fiber diet in mice led to an irreversible extinction of bacteria that depend on fiber for their metabolism [68] as well as a sensitized microbiome, where mice fed a fiber-free diet and then exposed to antibiotics showed a collapse of their bacterial microbiome with slower recovery as compared to their healthy counterparts [74]. While the phage dynamics in response to a low-fiber diet has not been directly assessed, some inferences can be made. Mucin-binding domains on phages have been identified [16,75], suggesting that the degradation of mucin in a low-fiber environment may change the nature of phage predation. Additionally, GI dysbiosis associated with a low-fiber diet can lead to increased reactive oxygen species (ROS) [76,77], which is theorized to trigger phage induction to enter the lytic cycle—where bacteriophages induce bacterial lysis for the release of phage progeny—thereby increasing phage abundance and predation [53,54].

### 3.3. Western Diet—High Sugar

The consumption of high sugar and other simple carbohydrates in the WD has been linked with adverse health outcomes. High sugar consumption is associated with the expansion of certain bacteriophage populations and with the development of metabolic diseases including Metabolic Syndrome (MS) and Type 2 Diabetes (T2D). High sugar consumption shifts the bacterial component of the microbiome toward a Pseudomonadota dominant phenotype with associated loss of Bacteroidota abundance [78,79] and detrimental effects on T-cell-specific immune-mediated protections [79]. These microbial changes are similar to those found in dysbiosis seen in metabolic disorders [80]. Removal of the high sugar component from a combinatorial high-fat, high-sugar diet in mice restores T-cell-mediated protections demonstrating specific effects by high sugar consumption [79]. Perhaps, unsurprisingly, given the microbial shifts associated with T2D, phage populations also expand in patients as assessed through fecal sequencing [81,82]. While early studies identified increased phage populations [82], improvements in sequencing and in databases for sequence comparison revealed that phages specific to *Enterobacteriaceae* are enhanced in T2D patients from China [81]. Given the interindividual diversity of the phageome, it is possible that T2D patients in other locations also undergo expansion of different phage populations. Surprisingly, in both these T2D studies, the bacterial hosts associated with the expanded phage populations showed no change in abundance as compared to controls, demonstrating that phageome abundances do not always mirror their bacterial counterparts and should be considered separately.

A more direct impact of sugar and sugar substitutes on phages was demonstrated in murine models [83,84]. Bacteria utilize a wide range of monosaccharide and disaccharide sugars (e.g., fructose, glucose, sucrose) to generate a variety of metabolic byproducts including lactic acid, ethanol, and acetic acid [85]. Administration of fructose to mice through supplementation of their drinking water resulted in increased production of phages specific to *Limosilactobacillus reuteri (L. reuteri)*; higher than that seen with glucose administration [83]. These shifts were concurrent with decreased *L. reuteri* survival. Fructose shifts the metabolic flux in *L. reuteri* such that there is an increased production of the short-chain fatty acid (SCFA) acetic acid, suggesting that either this metabolic shift or the increased concentration of acetic acid resulted in prophage induction with subsequent increased abundance of “free” *L. reuteri*-specific phages. Boling et al. [84] demonstrated that a variety of sugar substitutes also caused prophage induction. Stevia was the most potent inducer of prophages from *Bacteroides thetaiotaomicron* and *Staphylococcus aureus,* and aspartame induced prophages from *Enterococcus faecalis*. These effects were only tested in vitro with these three bacterial strains but indicate the direct impact sugar and sugar alternatives may have on the intestinal phageome.

### 3.4. Phage and Western Diet

When taken together—the results from a variety of murine models fed diets to mimic the WD—bacteriophage effects appear to be mixed and dependent on the element (fat, sugar, or fiber) being tested. Low fiber is likely to drive bacterial shifts first through the loss of a crucial food source and modification of the GI environment (i.e., loss of mucin) such that the combined hit drives the loss of bacterial populations critical for the maintenance of homeostasis and opens niches for expansion of pathobionts. In this instance, expansion of bacteriophage populations would occur upon prophage induction due to bacterial stress (increased ROS) or molecules associated with the changed bacterial dynamics (changes to the abundances and types of autoinducers). Results from examination of high-fat and high-sugar diets suggest that bacteriophage populations may shift first with follow-on effects to the bacterial abundances. The effects of fat and sugar on bacterial metabolism are likely less impactful, or work more slowly, than the loss of fiber, such that prophage induction occurs more quickly, driving up phage abundances, and this, in turn, results in shifts in the surrounding bacterial populations. This is a supposition on the authors’ part and would further require examination of the phage types and abundances throughout the administration of WD, especially early on, in order to track the phage–bacteria dynamics in response to dietary changes.

### 3.5. Specialized Diets

In recent years, a variety of specialized diets have gained attention and notoriety with claims of promoting health. These include the keto diet, gluten-free diet (GFD), the Atkins diet, and the Mediterranean diet among others. Several studies have described microbial shifts that accompany these diets [86,87,88,89,90,91,92]. However, comparably little research has been conducted to assess the effects of these diets on the phageome. With several studies demonstrating shifts in GI bacterial populations in response to gluten-free or low-gluten diets [93,94,95], Garmaeva et al. [96] examined the fecal virome of patients undergoing 4 weeks of a GFD followed by a 5-week washout period to characterize the effects on the phageome. In addition to confirming findings from Shkoporov et al. [21] demonstrating the individual specificity of the phageome, they showed that individuals with lower initial viral diversity had associated larger changes in the virome upon dietary intervention. As these results were from healthy individuals given a GFD, it is possible that shifts in bacteriophage populations would differ drastically in individuals with gut-related complications, such as Celiac disease, when changing to a GFD. Plant-based diets, such as vegetarian and vegan diets, support the development of a more diverse and stable gut microbiome [97] and are associated with higher proportions of Bacteroidota as compared to omnivorous diets [98]. This is exemplified in a study by Zuo et al. examining the fecal viromes of individuals from the Hong Kong (urban) and Yunnan (rural) regions in China [49]. Zuo et al. examined the contributions of a variety of factors to the overall composition of the gut virome including geography, urbanization, and consumption of ethnic-specific diets. In keeping with other previously reported findings, Zuo et al. also found the phageome to be highly individualistic, but further demonstrated that geography and urbanization were key factors for this inter-individualism. However, examination of diet between urban and rural groups demonstrated its own significant impact on the gut DNA virome with plant-based dietary factors (consisting of a variety of fruits and Chinese cabbage) acting as the highest contributors. For example, ethnic Zang individuals were found to have a higher abundance of *Bacteroides* phages. This matches their ethnic-specific diet, abundant in polysaccharides from barley and buttermilk tea, which supports the *Bacteroides* bacterial hosts that utilize these polysaccharides. In this same study, more associations between dietary components (such as fruits, tea, vegetables, etc.) and gut-DNA virome variation were identified than between these same components and gut-bacteriome variation. This same trend held true in correlations between dietary components and virus species vs. bacterial species, suggesting that diet has a greater impact on the phageome than on the bacteriome.

With many of these specialized diets, the measured effects on the composition of GI microbiota indicate shifts in many members [99], indicating the complexity involved in sorting out the full contributive effects that the different elements (dietary, microbial, or viral) may have on the system. However, one diet offers a unique opportunity to potentially investigate fluctuations in the phageome in response to dietary changes. One of the treatments for Irritable Bowel Syndrome (IBS) is the use of a low FODMAP (Fermentable Oligo-, Di-, Mono-saccharides, And Polyols) diet. This diet entails restriction of FODMAPS to help alleviate IBS symptoms [100]. However, even with the substantial change in dietary nutrients, measure of fecal microbiome composition, short-chain fatty acids (SCFAs), branched-chain fatty acids, and pH showed no substantial changes between pre- and post-administration of the diet [101,102]. The only consistent change seen in several clinical studies was the decrease in *Bifidobacterium* after diet administration [100,102]. Given the importance of the *Bifidobacterium* population on GI health [103], and its carriage of several prophages [52,104], the shift in this particular population offers the opportunity to better understand how nutrient deprivation may drive prophage release and how fluctuations in phages may drive the depletion of a single microbial population.

### 3.6. Effects of Malnutrition

While the WD represents overconsumption of dietary components (sugar and fat), the opposite of this, malnutrition, represents the effects of underconsumption, which also affects the microbiota of the GI system. Malnutrition is a dietary deficiency of calories or nutrients and is a serious source of morbidity and mortality, especially in children [105]. The bacteriome of childhood malnutrition is associated with low alpha diversity and disproportionate expansion of Pseudomonadota [106]. Furthermore, the microbiome maintains an “immature” state as postnatal maturation of the gut microbiota is perturbed [106,107]. Dysbiosis of malnutrition also influences the phageome and is transferrable to gnotobiotic mice through fecal viral transfer causing weight loss, barrier disruption, and metabolic phenotypes [107,108,109]. Treatment with antibiotics concurrent with the provision of nutritious food has shown better outcomes and faster recovery of a healthy microbiome [110,111,112]. However, the ready-to-use therapeutic food interventions alone in child populations suffering from severe acute malnutrition are transient [106,107]. While gut bacteria have been shown to be causally related to malnutrition [107,109], the role of the virome is less well understood. Reyes et al. [108] sequenced the fecal viromes of discordant and concordant Malawian twins suffering from malnutrition finding that viromes of concordant twins underwent age-related changes with perturbations that were similar in malnourished singleton individuals and discordant twins. These findings suggested a specific pattern of virome abnormality linked to malnutrition. Furthermore, several viral contigs associated with a malnourished state were identified with some of these same viral contigs (associated with *Caudoviricetes* and *Inoviridae* phages) identifiable in diseased gnotobiotic mice following fecal viral transfer from human subjects. Khan Mirzaei et al. [40] also identified a perturbed phageome in malnourished children with increased abundances of *Caudoviricetes* (especially those previously identified as *Siphoviridae*) correlating to the increase in Pseudomonadota.

### 3.7. Malnutrition vs. Overnutrition and Obesity

Overnutrition that causes diet-induced obesity has been shown to alter the phage community composition and its distribution in the gut. As Kim et al. demonstrated, mucosal and luminal phage density was increased with diet-induced obesity through the administration of a HFHS diet model in mice [63]. It is hard to tease out the effect of excess calories itself, since most overnutrition models involve the provision of high fat and high sugar, which we have reviewed in the above sections. For instance, a greater degree of enrichment in *Caudoviricetes* phages is observed within the colonic mucosa as compared to the lumen in obese mice given a HFHS diet. In these diet-induced obese mice, changes involving phage diversity were reversed when the animals were subsequently fed a low-fat diet, indicating that the dietary composition was the key factor in driving changes in the phages [63].

### 3.8. Effects of Plant Products

In addition to specific diets, nutritional elements on their own have been shown to have phage-related effects in specific ways: through induction/suppression of prophage or through a reduction in phage infectivity of its bacterial host (see Table 1 for a detailed list; modified from [113]). Most of the plant products defined as regulating phages are classified as polyphenols, which can be further broken into the subgroups of flavonoids and phenolic acids. Foods high in polyphenols have been shown to have several health benefits (reviewed in [114]), thus their consumption is recommended. Polyphenols are structurally diverse, naturally occurring phenols that are abundant in plants, thus introduced to the GI tract via the consumption of foods derived from plants.

Most research on the effects of plant products on the phageome has been carried out in vitro through the addition of a specific element to cultures of a phage and its corresponding host bacteria, with an examination of the outcomes of the activity of phage–bacteria interactions (reviewed in [54,113]). While informative as to nutritional supplements that may provide potential therapies or mechanisms of modulating the phageome, these findings remain to be translated in vivo. However, there are studies carried out in murine models demonstrating prophage induction within the intestine by dietary supplementation. For example, Boling et al. [84] showed prophage activation in a response to a variety of supplements (clove, artificial sweeteners, grapefruit seed extract, and propolis).

### 3.9. Effects of Short-Chain Fatty Acids

SCFAs are the metabolic byproducts of dietary fiber fermentation by the GI bacteria within the distal small intestine, cecum, and colon [139]. These one-to-six carbon, saturated aliphatic organic acids are an essential waste product of metabolism, not only to balance redox equivalent production in the anaerobic gut but also to maintain a healthy distribution of GI microbiota and epithelial barrier strength (reviewed in [140]). Acetate, propionate, and butyrate are the dominant SCFA and occur at a 60:20:20 proportion within the colon and stool [141,142,143,144]. Changes in SCFA levels are associated with leaky gut (barrier disruption) and development of GI diseases, such as Inflammatory Bowel Disease (IBD; [145]).

In 2019, Oh et al. [83] demonstrated that the incorporation of SCFA into the growth media of *Limosilactobacillus reuteri* (*L. reuteri*) resulted in the activation of prophage. Acetic acid, butyric acid, and propionic acid—but not lactic acid—all resulted in increased phage production with no effect on bacterial growth. Less direct evidence for the ability of SCFA to induce prophage can be found in vivo and in vitro. Providing more substrate for bacterial fermentation through high-fiber diets has been shown to increase prophage production [146]. Although increases in dietary fiber have not always been shown to lead to subsequent increases in SCFA [147], it is the main source of SCFA production. In vitro, the addition of acetic acid to *Staphylococcus aureus* cultures induced a phage-encoded enterotoxin (SEA) [148]. Although a full bacteriophage was not produced in this result, the controlling mechanisms of induction were phage-derived, suggesting a larger population of prophages is potentially regulated by SCFA concentrations.

## 4. Gut Phageome Effects on Disease

Digestive diseases are closely linked to the health and activity of the gut microbiome, and even though the gut phageome needs to be studied in greater depth, there are already several reports characterizing phageome shifts in patients with digestive diseases. These include *Caudoviricetes* expansion in patients with IBS, and decreased abundance and diversity of phages such as *Spounaviridae* (part of what were previously classified as *Myoviridae*), which prey on *Clostridiales*, in patients with colitis [149,150]. Increases in *Microviridae* and phages previously classified as *Podoviridae* have been found in subjects with IBS with constipation (IBS-C) and IBS with diarrhea (IBS-D), with additional increases in phages previously classified as *Siphoviridae* or *Myoviridae* in IBS-C and IBS-D, respectively [151]. These described shifts in phage populations have the potential to affect therapeutic strategies. For example, in 2012, a clinical trial with IBS patients (diagnosed according to the Rome III criteria) examined the effects of probiotic administration on IBS patients. Murakami et al. found that the administration of plant-derived *Levilactobacillus brevis* (*L. brevis* KB290) alleviated IBS symptoms, increased fecal frequencies of the genera *Bifidobacterium* while decreasing those of the genera *Clostridium*, and increased quality-of-life scores [152]. However, the subsequent paper by Mihindukulasuriya et al. [151] found subjects with IBS-C specifically had increased levels of free (not residing within the bacterial host) lysogen LBR-48, a phage targeting *Levilactobacillus brevis*. The presence of increased levels of this particular phage within the colonic luminal environment suggests that administration of the *L. brevis* KB290 probiotic in this subset of IBS patients might not be successful due to phage predation by LBR-48. Thus, successful treatment of diseases, such as IBS, may instead require a more tailored approach with the phageome of the patient considered prior to the administration of probiotics. Various disease states that change the microbial ecology also alter phage diversity, suggesting a potential target of treatment may involve the provision of beneficial bacteria (probiotics) along with a matching phage population. Moreover, the success of fecal microbial transplants (FMTs) for *Clostridioides difficile* (*C. difficile*) infection may not require the transfer of bacteria to obtain a beneficial response to treatment. Transfer of bacteria-free fecal filtrate containing viral components (fecal virome transplant; FVT) reduced symptoms of *C. difficile* more than traditional FMT [153], suggesting that therapeutic effects may derive from the transfer of the phageome. Similar findings have demonstrated phageome regulation of obesity with transplants of the fecal virome (dominated by bacteriophage) from slim mice to HFD-fed obese mice resulting in a significant decrease in weight and improved symptoms related to T2D [154].

Phageome effects have been found to also extend beyond the GI tract. Subjects with alcohol-associated liver disease have been shown to have increases in *Enterobacteria* phages and *Escherichia* phages [155]. In contrast, subjects with non-alcoholic fatty liver disease (NAFLD), which correlates with alterations in the gut microbiome, have decreased intestinal viral diversity [156]. In these patients, increased disease severity was inversely correlated to the proportion of bacteriophages compared to other intestinal viruses.

## 5. Opportunities for Future Research

With the complexity of the gut phageome and its correlation to human health, there are numerous areas of research to expand our knowledge. With recent improvements in sequencing and bioinformatics, additional longitudinal studies would better define the phages of a healthy gut across the lifespan. Now that more detailed identification of intestinal phages is possible, a revisitation of the abundance and types of phages within the human virome longitudinally throughout human development can occur. Techniques such as proximal ligation [157,158,159] could illuminate the bacteria–phage interactions in a contextual manner to allow the identification of healthy interactions vs. those correlated to different diseases. Focus on characterizing the lytic and lysogenic phases of phages and what triggers may activate/inactivate each state within the GI tract would inform disease triggers as well as potential therapies. While biorepositories for phages have expanded, more gut-associated samples will allow for research to focus on a variety of correlations that may exist between the phageome and health of the individual as well as specific populations, both bacterial and viral. Furthermore, translational animal studies and large randomized-control trials can help further clarify the effects of the gut phageome on human immunity.

As compared to research on the effect of diet on the bacterial microbiome, much less is known about the specific dietary effects on bacteriophage populations. Even fewer studies have examined the effects the phage component associated with certain foods (i.e., fermented foods) might have on the GI phageome, the rest of the microbiome, and the host. As we previously mentioned, there are two ways in which diet may affect the phageome, either indirectly (affecting the bacterial hosts on which phage prey) or directly (affecting phage induction or activity). As current evidence suggests that diet works through both mechanisms, identifying phage changes occurring within specific diets, such as vegetarian or vegan, would help researchers understand phageome flux in response to dietary intake. With this, it would be beneficial to explore the ideas of longitudinal studies to confirm if dietary disturbances could permanently change phage diversity. More studies involving bacteriome–phageome interactions with environmental factors such as diet will help solidify the full picture of collaboration and/or competition between gut microbiota and phageome.

## 6. Conclusions

The possibility of modulating the human phageome through dietary interventions remains an intriguing therapeutic alternative. In vivo studies have already demonstrated that the expansion of phage populations is associated with health [97] and that phageome shifts are concurrent with dysbiosis [63,112]. As such, investigation into dietary elements that can stimulate the production of or increases in populations of bacteriophages associated with a healthy phageome and translation of these in vitro results may also offer promising options to a variety of conditions (reviewed in [53,54]) beyond traditional phage therapy approaches.

## Figures and Tables

**Table 1 ijms-25-08690-t001:** Effects of plant products on bacteriophage behavior.

Effect	Nutrient	Class	Phage	Reference
Prophage Induction	Caffeic acid	Phenolic acids	λ	[115]
	Propolis	Flavonoids	Unreported	[84]
	Caffeine	Alkaloids	φX174	[116]
	GCG	Flavonoids	933 W	[117]
Prophage Suppression	Carvacrol, thymol	Phenolic acids	933 W	[118]
	EGCG	Flavonoids	933 J	[117]
	Propolis	Flavonoids	Unreported	[84,117]
	GSE	Flavonoids	933 W	[119]
	Cinnamaldehyde (cinnamon)	Essential oil (aldehydes)	933 W	[120,121]
	Oregano	Essential oil	Unreported	[84]
Infectivity Reduction	Gallic acid, chlorogenic acid	Phenolic acids	PL-1, MS2, Av-5	[122,123]
	Caffeic acid	Phenolic acids	Av-5, MS2	[123]
	Tea extracts	Phenolic acids or flavonoids	Felix 01, P22	[124]
	Pomegranate juice	Phenolic acids or flavonoids	MS2	[125]
	Catechins	Flavonoids	T4	[126]
	Cranberry juice	Flavonoids	T2, T4	[127]
	Red propolis (formononetin)	Flavonoids	MS2, Av-08	[128]
	Chamomile, lemongrass, cinnamon	Essential oils	T7, SA	[129]
	Proanthocyanidin	Flavonoids	MS2, φX174	[125]
	PJE	Flavonoids	MS2	[130]
	Ascorbic acid	Vitamin	φX174, δA, T7, P22, D29, PM2, MS2	[131,132,133,134,135,136]
	GSE	Flavonoids	MS2	[137]
	Psoralen	Furocoumarins	MS2	[138]

Abbreviations: Epigallocatechin Gallate (EGCG); Gallocatechin-3-gallate (GCG); grape seed extract (GSE); *Petasites japonicus* extract (PJE).

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
