# Peer review of "Dietary Effects on the Gut Phageome"

_ijms, 2024, doi:10.3390/ijms25168690_

Round 1

Reviewer 1 Report

Comments and Suggestions for Authors

This study describes the effect of dietary habits on the gut phagome-bacteriophages living amongst and within our bacterial microbiome. The presented study is valuable as it shifts from the very deeply studied microbiome to less studied phageome.

In order to improve this version of the manuscript my recommendations are:

1. Lines 75, 76, and 159 ad a space before references (42,43), (21,44-47), (76,77) respectively.

2. Line 160 -please explain briefly the process of phage lytic cycle.

3. Line 188- please write Limosilactobacillus reuteri instead of L. reuteri

4. lLine 195 please write full name of microorganisms B. thetaiotaomicrom, S. aureus and E. faecalis.

5. Line 290- number of subtitle is not write-please replace 3.7 with 3.8

6. Line 311- number of subtitle is not write-please replace 3.7 with 3.8

7. Table 1. it is not clear from the first part of the table if prophage has induction or repression effect? Please describe which effect prophage posses.

8. Line 352- Novel nomenclature is Clostridioides difficile please change the name of the bacteria in the text. Also replace C. diff with C. difficile.

9. Line 357: What VLPs stands for- Virus Like Particles? Please explain abbreviation.

Author Response

The authors would like to thank this reviewer for taking the time to critically assess this review. In recognition of the subsequent suggestions, the authors have made the following changes.

  1. Lines 75, 76, and 159 ad a space before references (42,43), (21,44-47), (76,77) respectively.

                Authors have corrected

  1. Line 160 -please explain briefly the process of phage lytic cycle.

            Authors have corrected

  1. Line 188- please write Limosilactobacillus reuteri instead of L. reuteri

            Authors have corrected

  1. lLine 195 please write full nameof microorganisms B. thetaiotaomicrom,S. aureus and E. faecalis.

            Authors have corrected

  1. Line 290- number of subtitle is not write-please replace 3.7 with 3.8

            Authors have corrected

  1. Line 311- number of subtitle is not write-please replace 3.7 with 3.8

            Authors have corrected

  1. Table 1. it is not clear from the first part of the table if prophage has induction or repression effect? Please describe which effect prophage posses.

            Authors have broken table into further subcategories for clarification

  1. Line 352- Novel nomenclature is Clostridioides difficileplease change the name of the bacteria in the text. Also replace C. diff with C. difficile.

            Authors have corrected

  1. Line 357: What VLPs stands for- Virus Like Particles? Please explain abbreviation.

            To avoid confusion, authors have changed this to “fecal virome”, line 383

Reviewer 2 Report

Comments and Suggestions for Authors

In recent years, several studies have explored the possible role of gut microbiota in many diseases. However, there is still a scarce knowledge regarding the role of viroma in the gut microbial ecosystem. Therefore, this review appears really interesting moreover because it sheds new light on the  current knowledge about the effects of the diet on the gut phageome.

Major revisions:

1.     Among the diets that have been gaining success in the last years due to their clinical efficacy in the treatment of gastrointestinal diseases is the low FODMAP diet (LFD). The authors could profitably use the  paper by Bellini M, Tonarelli S, eta al  A. Low FODMAP Diet: Evidence, Doubts, and Hopes. Nutrients. 2020 Jan 4;12(1):148. doi: 10.3390/nu12010148. PMID: 31947991; PMCID: PMC7019579.) in section “3.5 Specialised diets”, as it is increasingly being suggested for the treatment of Irritable Bowel Syndrome (IBS), a disorder which the authors further discussed. The authors should highlight the possible dysbiotic effects of LFD on the gut microbiota

2.     Lines 342-348: The authors report the results of two studies [145-146] showing that the strain L. brevis KB290 is effective to increase quality of life scores in a probiotic trial with IBS patients, on the other hand an elevated abundance of Lactobacillus virus LBR48, a phage targeting L. brevis, could make this probiotic treatment less effective. Therefore, Mihindukulasuriya et al. [145] concluded by asking whether or not probiotic therapies should be tailored to the phage population present in an individual's gut. the authors should better elucidate the contents expressed in these two studies [145-146] in order to make them easily accessible to any reader.

Minor revisions:

1.     There are some typos errors to be corrected: lines 97, 158, 203, 255, 299, 313, 316, 320, 359 (use the contracted form GI); line 114, line 143, line 163, line 201, line 215, line 253 (use the contracted form WD); line 312 (use the contracted form SCFAs); line 341 (use the extended form constipation and diarrhea respectively); line 357 (use the extended form virus-like particles); line 358 (use the contracted form T2D).

2.     Concerning Table 1: I would suggest to the authors adding at the bottom of the table the abbreviations of all acronyms used (i.e. EGCG, GCG, GSE and PJE).

Comments on the Quality of English Language

In recent years, several studies have explored the possible role of gut microbiota in many diseases. However, there is still a scarce knowledge regarding the role of viroma in the gut microbial ecosystem. Therefore, this review appears really interesting moreover because it sheds new light on the  current knowledge about the effects of the diet on the gut phageome.

Major revisions:

1.     Among the diets that have been gaining success in the last years due to their clinical efficacy in the treatment of gastrointestinal diseases is the low FODMAP diet (LFD). The authors could profitably use the  paper by Bellini M, Tonarelli S, eta al  A. Low FODMAP Diet: Evidence, Doubts, and Hopes. Nutrients. 2020 Jan 4;12(1):148. doi: 10.3390/nu12010148. PMID: 31947991; PMCID: PMC7019579.) in section “3.5 Specialised diets”, as it is increasingly being suggested for the treatment of Irritable Bowel Syndrome (IBS), a disorder which the authors further discussed. The authors should highlight the possible dysbiotic effects of LFD on the gut microbiota

2.     In lines 342-348 the authors report  the results of two studies [145-146] showing that the strain L. brevis KB290 is effective to increase quality of life scores in a probiotic trial with IBS patients, on the other hand an elevated abundance of Lactobacillus virus LBR48, a phage targeting L. brevis, could make this probiotic treatment less effective. Therefore, Mihindukulasuriya et al. [145] concluded by asking whether or not probiotic therapies should be tailored to the phage population present in an individual's gut.  The authors should better elucidate the contents expressed in the two studies [145-146] in order to make them  easily accessible to any reader.

Minor revisions:

1.     There are some typos errors that should be corrected: lines 97, 158, 203, 255, 299, 313, 316, 320, 359 (use the contracted form GI); line 114, line 143, line 163, line 201, line 215, line 253 (use the contracted form WD); line 312 (use the contracted form SCFAs); line 341 (use the extended form constipation and diarrhea respectively); line 357 (use the extended form virus-like particles); line 358 (use the contracted form T2D).

2.     Concerning Table 1: I would suggest to the authors adding at the bottom of the table the abbreviations of all acronyms used (i.e. EGCG, GCG, GSE and PJE).

Author Response

The authors would like to thank this reviewer for taking the time to critically assess this review. In recognition of the subsequent suggestions, the authors have made the following changes.

Major revisions:

  1. Among the diets that have been gaining success in the last years due to their clinical efficacy in the treatment of gastrointestinal diseases is the low FODMAP diet (LFD).The authors could profitably use the  paper by Bellini M, Tonarelli S, eta al  A. Low FODMAP Diet: Evidence, Doubts, and Hopes. Nutrients. 2020 Jan 4;12(1):148. doi: 10.3390/nu12010148. PMID: 31947991; PMCID: PMC7019579.) in section “3.5 Specialised diets”, as it is increasingly being suggested for the treatment of Irritable Bowel Syndrome (IBS), a disorder which the authors further discussed. The authors should highlight the possible dysbiotic effects of LFD on the gut microbiota

                The authors have added a section on the low FODMAP diet in IBS patients to section 3.5 (lines 253-268) focusing on the use of this diet for future phageome research.

  1.    Lines 342-348: The authors report the results of two studies [145-146] showing that the strain L. brevis KB290 is effective to increase quality of life scores in a probiotic trial with IBS patients, on the other hand an elevated abundance of Lactobacillus virus LBR48, a phage targeting L. brevis, could make this probiotic treatment less effective. Therefore, Mihindukulasuriya et al. [145] concluded by asking whether or not probiotic therapies should be tailored to the phage population present in an individual's gut. the authors should better elucidate the contents expressed in these two studies [145-146] in order to make them easily accessible to any reader.

            The authors have expanded the information provided from both of these references and on the relation between them to highlight the consideration of a patients phageome for probiotic treatment (lines 361-374).

Minor revisions:

  1. There are some typos errors to be corrected: lines 97, 158, 203, 255, 299, 313, 316, 320, 359 (use the contracted form GI); line 114, line 143, line 163, line 201, line 215, line 253 (use the contracted form WD); line 312 (use the contracted form SCFAs); line 341 (use the extended form constipation and diarrhea respectively); line 357 (use the extended form virus-like particles); line 358 (use the contracted form T2D).

                Authors have made all of these corrections

  1. Concerning Table 1: I would suggest to the authors adding at the bottom of the table the abbreviations of all acronyms used (i.e. EGCG, GCG, GSE and PJE).

            Authors have added these abbreviation descriptions

Round 2

Reviewer 2 Report

Comments and Suggestions for Authors

The authors fully answered all the previously raised questions comments 

Comments on the Quality of English Language

see above